# In Vitro/In Vivo Translation of Synergistic Combination of MDM2 and MEK Inhibitors in Melanoma Using PBPK/PD Modelling: Part II

**DOI:** 10.3390/ijms231911939

**Published:** 2022-10-08

**Authors:** Jakub Witkowski, Sebastian Polak, Zbigniew Rogulski, Dariusz Pawelec

**Affiliations:** 1Faculty of Chemistry, University of Warsaw, Pasteura 1, 02-093 Warsaw, Poland; 2Adamed Pharma S.A., Adamkiewicza 6a, 05-152 Czosnów, Poland; 3Faculty of Pharmacy, Jagiellonian University, Medyczna 9, 30-688 Krakow, Poland; 4Simcyp Division, Certara UK Limited, Level 2-Acero, 1 Concourse Way, Sheffield S1 2BJ, UK

**Keywords:** anticancer drugs, preclinical study, pharmacokinetics, pharmacodynamics, drug combination, PBPK/PD modelling, MDM2 inhibitor, MEK inhibitor

## Abstract

The development of in vitro/in vivo translational methods for synergistically acting drug combinations is needed to identify the most effective therapeutic strategies. We performed PBPK/PD modelling for siremadlin, trametinib, and their combination at various dose levels and dosing schedules in an A375 xenografted mouse model (melanoma cells). In this study, we built models based on in vitro ADME and in vivo PK/PD data determined from the literature or estimated by the Simcyp Animal simulator (V21). The developed PBPK/PD models allowed us to account for the interactions between siremadlin and trametinib at PK and PD levels. The interaction at the PK level was described by an interplay between absorption and tumour disposition levels, whereas the PD interaction was based on the in vitro results. This approach allowed us to reasonably estimate the most synergistic and efficacious dosing schedules and dose levels for combinations of siremadlin and trametinib in mice. PBPK/PD modelling is a powerful tool that allows researchers to properly estimate the in vivo efficacy of the anticancer drug combination based on the results of in vitro studies. Such an approach based on in vitro and in vivo extrapolation may help researchers determine the most efficacious dosing strategies and will allow for the extrapolation of animal PBPK/PD models into clinical settings.

## 1. Introduction

Metastatic melanoma is a cancer condition that is dangerous and difficult to treat due to its ability to spread early and aggressively. Before the development of new therapeutic strategies, the median survival of patients with metastatic melanoma was only 6–9 months [1] and the 10-year survival rate was less than 10% [2]. Although recent therapeutic advances for metastatic melanoma have considerably increased the overall survival of patients with melanoma, a subset of patients do not respond to immunotherapy or targeted therapies [3,4]. Such limited responses may be explained by arising resistance. Drug combinations targeting multiple signalling pathways in cancer cells may provide a remedy for emerging resistance development [5], and this is why new anticancer drug combinations and therapies are so important and urgently required. One of the novel therapeutic options is the drug combination of siremadlin (MDM2 inhibitor) and trametinib (MEK inhibitor). Preclinical evidence suggests that siremadlin (previously known as HDM201) and trametinib synergistically act in melanoma treatment [6,7]. To assess how polytherapy may improve the anticancer response, the performance of preclinical translational studies and the development of in vitro/in vivo translational methods are highly needed. A bench-to-bedside approach is much more challenging for drug combinations than for a single drug because it must combine assumptions regarding the interaction between two (or more) drugs at both the pharmacokinetic (PK) and pharmacodynamic (PD) levels [8,9]. One of the solutions allowing for such a prediction is physiologically based pharmacokinetic–pharmacodynamic (PBPK/PD) modelling. The PBPK/PD modelling approach allows researchers to combine information on the drug characteristics with their knowledge of physiology and biology at the organ and whole-organism levels. Such an approach allows researchers to achieve a representation of the drug in a biological system and the simulation of drug concentration–time profiles (pharmacokinetic profiles) and to link it to the drug’s efficacy (its pharmacodynamic effect). This modelling approach offers an advantage over the traditional PK/PD modelling approach because it potentially allows for extrapolation into conditions for which pharmacokinetic studies have not been conducted. PBPK models consider different organs and tissues (whole-body PBPK model) that are the most relevant to the absorption, distribution, metabolism, and excretion (ADME) of the drug. Thus, the drug concentration–time (pharmacokinetic) profile can be accurately simulated in particular organs and tissues. Such a prediction is of high pharmacological relevance because it enables the estimation of drug exposure at the site of its action (for example, in a tumour), which may be difficult or impossible to experimentally measure in animals or humans.

Analysis of the interactions between two or more drugs at the PD level is difficult because of a lack of consensus on which theoretical model should be used to describe the drug interaction type. As previously discussed, this issue is usually related to a dilemma when a drug combination is classified as synergistic according to one model but antagonistic in the other [6,10,11,12]. In this study, we chose the previously proposed synergy metrics δ score (from the Synergyfinder package analysis) and β parameter (from the Synergy package analysis) to be tested as translational in vitro/in vivo PD interaction parameters [6].

Our main goal in this study was to develop and optimise a PBPK/PD model which could allow the translation of the in vitro drug combination results to an in vivo situation. An approach that involves using the results from in vitro studies coupled with PBPK/PD modelling may accurately describe the observed tumour growth inhibition (TGI) data and may suggest the most synergistic and efficacious schedules and dose levels for siremadlin and trametinib in mice in vivo. This modus operandi may lead to more accurate estimations of drug combination efficacy in virtual clinical trials (VCTs), which might be performed on a virtual representation of cancer patients and ultimately provide the rationale for using this drug combination in clinical trials on real patients with melanoma cancer.

## 2. Results

### 2.1. PBPK Models (with and without PK Interaction)

The developed PBPK models properly described the observed concentration–time data for siremadlin, trametinib, and their combination in plasma, A375 tumour, and other tissues (the muscle, spleen, brain, heart, kidneys, skin, lungs, gut, and liver), as shown in Figure 1, Figure 2, Figure 3, Figure 4, Figure 5, Figure 6 and Figure 7. These results are in line with those of a numerical analysis of fold errors (the predicted and observed values) for the most important pharmacokinetic parameters *AUC*_0–24h_, *C*_max_, and *T*_max_, which, in most cases, were within the 2-fold error range 0.5–2.0 (Appendix A).

Regarding the estimated *AUC*_0–24h_, the PBPK simulation for siremadlin indicated that this parameter was accurately predicted for all tissues within a fold error of 0.93, except for the A375 tumour tissue, which was slightly overestimated (1.22), as shown in Appendix A. We obtained a similar observation for the estimated trametinib *AUC*_0–24h_. The estimated fold errors were also close to unity (0.93) in all tissues except for the A375 tumour, for which we noted a 0.96-fold error (Appendix A).

Concerning the predicted *C*_max_ for siremadlin, this parameter was overpredicted in the following tissues: the plasma (1.10), A375 tumour (1.00), brain (1.31), skin (1.57), and gut (1.15); please refer to Figure 1, Figure 3a,c, Figure 6c and Figure 7c and Appendix A. Compared with the tissues that were well supplied with blood, the *C*_max_ was somewhat underpredicted in the muscles (0.89), spleen (0.95), heart (0.89), kidney (0.97), lungs (0.99), and liver (0.94), as shown in Figure 4a,c, Figure 5a,c, Figure 6a and Figure 7a and Appendix A. Regarding trametinib, the estimated fold errors for this PK parameter were somewhat overpredicted in all the tissues (range: 1.00–2.08), as shown in Figure 2, Figure 3b,d, Figure 4b,d, Figure 5b,d, Figure 6b,d and Figure 7b,d and Appendix A. We noted a significant fold error (2.08) in the brain tissue, which could have been caused by the suboptimal blood perfusion for this tissue in the constructed PBPK model (Figure 3b).

Regarding the *T*_max_ parameter for siremadlin, the estimations for this parameter were generally overpredicted for all the tissues (1 < fold error < 2). The exceptions were the gut and A375 tumour tissue, for which this parameter was slightly underpredicted (0.74) and remarkably overpredicted (4.26), respectively. This may have been caused by unoptimised tumour blood perfusion in the mouse model; alternatively, the observed T_max_ values might not have been not properly determined due to sparse sampling, as depicted in Figure 1, Figure 3a,c, Figure 4a,c, Figure 5a,c, Figure 6a,c and Figure 7a,c and Appendix A. Concerning the predicted *T*_max_ for trametinib, the calculated fold errors for most of the tissues were within a 2-fold error range. The only exception was for the brain tissue (fold error 0.11), which may have been caused by suboptimal blood perfusion for this tissue in the constructed PBPK model (Figure 2, Figure 3b,d, Figure 4b,d, Figure 5b,d, Figure 6b,d and Figure 7b,d and Appendix A).

Basic PBPK models were further improved by introducing PK interactions as a result of the coadministration of the two studied drugs. The PK interactions could be explained by the altered absorption process and distribution in the tumour compartment as well (Appendix A). The modification of those parameters allowed us to accurately fit the models to the data with PK interaction in the plasma and other organs (Figure 1, Figure 2, Figure 3, Figure 4, Figure 5, Figure 6 and Figure 7).

In the developed models assuming PK interactions, we estimated the most important pharmacokinetic parameters *(AUC*_0–24h_, *C*_max_, and *T*_max_). In most cases, those parameters were within the 2-fold error range (0.5–2.0). Regarding the estimated *AUC*_0–24h_ for the siremadlin model assuming PK interaction, this parameter was slightly underestimated for all the tissues (within a fold error range of 0.76–0.86), except for the A375 tumour tissue, for which this parameter was slightly overestimated (1.54), as shown in Appendix A. Our observation was similar for the estimated trametinib model assuming PK interaction. The estimated fold errors for the *AUC*_0–24h_ were also slightly underestimated (fold error range: 0.76–0.88) in all the tissues except for the A375 tumour and plasma, for which we noted 1.22- and 1.02-fold errors, respectively (Appendix A).

Concerning the predicted *C*_max_ for the siremadlin model assuming PK interaction, this parameter was overpredicted in the following tissues: the plasma (1.15), A375 tumour (1.00), brain (1.02), skin (1.31), and gut (1.00); please refer to Figure 1, Figure 3a,c, Figure 6c and Figure 7c and Appendix A. Compared with the tissues that were well supplied with blood, the *C*_max_ was somewhat underpredicted in the studied tissues, including the muscle (0.89), spleen (0.90), heart (0.94), kidney (0.84), lungs (0.86), and liver (0.86), as shown in Figure 4a,c, Figure 5a,c, Figure 6a and Figure 7a and Appendix A. Regarding the trametinib model considering PK interaction, the estimated fold errors for this parameter were slightly overpredicted in the following tissues: the plasma (1.10), A375 tumour (1.00), spleen (1.33), and brain (1.29). In the well-perfused tissues, the *C*_max_ was slightly underestimated in the muscle (0.88), heart (0.70), kidney (0.96), lungs (0.61), gut (0.77), and liver (0.85). However, in this model, we also noted that the *C*_max_ value of the skin was underestimated (Figure 2, Figure 3b,d, Figure 4b,d, Figure 5b,d, Figure 6b,d and Figure 7b,d and Appendix A).

Regarding the *T*_max_ parameter for the siremadlin model assuming PK interaction, estimations of this parameter was generally overpredicted for all the tissues (1 < fold error < 2), except for the liver and A375 tumour tissues, for which this parameter was slightly underpredicted (0.81) and considerably overpredicted (3.38), respectively. This may have been caused by unoptimised tumour blood perfusion in the mouse model, or the observed T_max_ values might not have been not properly determined due to sparse sampling, as depicted in Figure 1, Figure 3a,c, Figure 4a,c, Figure 5a,c, Figure 6a,c, Figure 7a,c and Appendix A. Concerning the predicted *T*_max_ for the trametinib model accounting for PK interaction, the calculated fold errors in most of the tissues were within a 2-fold error range. The only exceptions were for the brain tissue (fold error 0.17), which may have been caused by suboptimal blood perfusion for this tissue, and the gut (fold error 2.79), which may have been due to one or more of the following reasons: the combined effect of the high modification of the ka parameter, sparse sampling of the gut tissue homogenate, or homogenisation of the initial fragment of the intestine where the trametinib concentrations more quickly appeared (Figure 2, Figure 3b,d, Figure 4b,d, Figure 5b,d, Figure 6b,d and Figure 7b,d and Appendix A).

Generally, T_max_ was the PK parameter most often mispredicted. Such a discrepancy between the predicted and observed values might have been related to the fact that we report the observed T_max_ values on the highest observed C_max_, which might not have been properly determined due to the sparse sampling of the observed data.

In the final stage of the siremadlin and trametinib PBPK model development, we compared the models’ predictions with external PK data digitised from the literature [13,14]. We assumed that differences in the used formulations between our study and the already-published data would only be associated with the absorption process (Appendix A). As shown in Appendix A, the developed PBPK models were able to effectively capture the plasma concentration–time data observed in the external studies, which therefore validated those models. The final PBPK models’ parameters with and without PK interactions are summarised in Appendix A.

### 2.2. PD (TGI) Models

To determine the characteristics of unperturbed tumour growth, we plotted the log tumour volume versus time for the mean tumour volume in vehicle-treated animals, as shown in Appendix A. As a result, the growth curve in the vehicle control group was initially characterised by a fast-growing exponential phase that ultimately approached a plateau once a certain tumour volume was reached, indicating the saturation of the tumour growth. According to our current knowledge, this may have been caused by rapid tumour growth, which leads to limited oxygen and nutrient supply [15]. The selected logistic growth model best described the unperturbed tumour growth in terms of the model score.

The final perturbed TGI models for siremadlin and trametinib assumed logistic tumour growth, the Skipper–Schabel–Wilcox (log-kill) tumour-cell-killing hypothesis, drug effects described by the exponential drug-killing model, acquired resistance to the therapy, and treatment effect delay described by the signal distribution model with four transit compartments. For drugs administered in monotherapy, those models were able to accurately capture the changes in tumour volume in time, with a mean relative error (RE) < 20%, as shown in Figure 8 and Appendix A.

To properly fit the tumour volume data, TGI modelling for drug combinations required additional parameters for the drug interactions at the PK and PD levels. We tested two different parameters that were determined to be drug interaction parameters at the PD level during the in vitro data analysis (Appendix A). We selected the β parameter from the MuSyC drug interaction model as the translational in vitro/in vivo PD parameter. We predicted that the interactions at the PK level (the AUC ratio parameters for siremadlin and trametinib) would be dose-dependent in all the tested drug combination arms. The predicted tumour volumes for the drug combination were within a mean RE of <20%, as shown in Figure 9 and Appendix A.

We successfully verified the TGI models with external efficacy data, as described in Section 4.5.2 and as shown in Appendix A.

The key input parameters for the final TGI models for the single drug and drug combination are shown in Appendix A.

### 2.3. PBPK/PD Estimation with Universal Model for Drug Combination at Human Equivalent Doses (HEDs)

We created a universal model based on a visual inspection of the data from the current study, validation studies, and score analysis. Determining the relationships between the TGI model parameters allowed us to carefully extrapolate the values of the model parameters for different doses and dosing regimens (Appendix A). As shown in Figure 10, the results of the tumour volume simulations after two cycles of the combined therapy at HEDs revealed that at least 40 days (960 h) of continuous dosing of trametinib with siremadlin (regardless of the siremadlin dosing regimen) is needed for complete tumour regression (tumour volume ≤ 32 mm^3^). Additionally, the results from simulations for two therapy cycles suggested that the synergistic efficacy of the siremadlin and trametinib drug combination reduces the number of trametinib doses needed to achieve tumour stasis (tumour volume ≤ 170 mm^3^) to only 21 doses in a treatment cycle. In general simulations of combinations accounting for continuous siremadlin dosing, the results indicated that qdx7 or qdx14 had a higher efficacy than intermittent dosing schedules qwx2 or qdx1, as shown in Appendix A. Simulations of siremadlin, trametinib, and their combination efficacy after one and two cycles of therapy at HEDs are depicted in Appendix A.

## 3. Discussion

The developed mouse PBPK/PD models for the MDM2 inhibitor siremadlin, the MEK inhibitor trametinib, and their therapeutic combination were able to describe both the pharmacokinetic and pharmacodynamic profiles of those drugs. The models consider the oral (P.O.) administration of siremadlin and trametinib, full-body distribution model, hepatic metabolism for siremadlin, and intravenous clearance for trametinib. Additionally, the models implement a permeability-limited tumour distribution model and drug interactions at absorption and tumour distribution levels, which allowed us to capture changes in siremadlin and trametinib concentrations in plasma and other tissues (e.g., the heart, liver, spleen, muscle, brain, kidney, A375 tumour, lung, gut, and skin) when we separately or simultaneously administered both compounds. The interactions at the absorption level include changes in the absorption rate constant (ka) for siremadlin and alterations in the absorption rate constant (ka), fraction absorbed (fa), and lag time (tlag) in the case of trametinib. We hypothesised that those two drugs are competing for intestinal transporters related to absorption. Whereas siremadlin might be preferentially transported, a saturation of absorption and transport mechanisms may decrease trametinib absorption. In turn, in tumour interaction depending on the shift in passive permeability and efflux P-gp transporter clearances, an increased disposition of both compounds may occur, which may additionally abolish tumour resistance, which is often related to increased efflux transporter abundance and activity [16,17,18,19,20,21]. However, further PK studies are needed to confirm such observations in plasma and tumour compartments. The application of PBPK models for siremadlin and trametinib allowed us to calculate key the pharmacokinetic parameters *AUC*_0–24h_, *C*_max_, and *T*_max_, which were mainly within the 2-fold error range for all tissues except the A375 tumour and brain tissues. This may have been caused by suboptimal blood perfusion in those tissues. In the case of the A375 tumour, a slower distribution and, consequently, a higher predicted *T*_max_ than that observed was probably related to insufficient blood perfusion. Applying higher blood perfusion would be possible, but it might be out of the range of measured perfusion in human melanoma xenografts [22,23,24]. In models assuming coadministration and associated PK interaction, we also characterised the *AUC*_0–24h_, *C*_max_, and *T*_max_ mainly within the 2-fold range for all the tissues except for the A375 tumour, brain, and gut. Regarding the A375 tumour and brain tissues, the high *T*_max_ fold error might be related to unoptimised blood perfusion in this model. The higher calculated trametinib *T*_max_ for the gut may have been related to the combined effect of the considerable modification of the ka parameter with the sparse sampling of the gut tissue homogenate, or to the homogenisation of the initial fragment of the intestine (where the concentrations more quickly appeared), which could have impacted the predicted *T*_max_ for this tissue. The T_max_ was the PK parameter most often mispredicted. Such a discrepancy between the predicted and observed values for this parameter might have been related to the fact that we used the observed T_max_ values on the highest observed C_max_, which might not have been accurately determined because of the sparse sampling of the observed data in this study. We successfully verified the obtained PBPK models with external PD data extracted from the literature [13,14].

The developed TGI models describe the time course of siremadlin and trametinib efficacy when administered separately and together to mice xenografted with A375 melanoma cells. The logistic growth model best described the unperturbed tumour growth using data from the current study and validation sets.

Based on previously performed in vitro studies, it is known that compounds’ killing effect is concentration- and time-dependent with an initial delay in the response, and resistance that might arise. These initial assumptions led us to develop final perturbed TGI models for siremadlin and trametinib. The models assume logistic tumour growth, the Skipper–Schabel–Wilcox (log-kill) tumour-cell-killing hypothesis, drug effect described by the exponential drug killing model, acquired resistance to the therapy, and treatment effect delay described by the signal distribution model with four transit compartments. The delay in the effect of these drugs is most likely related to the duration of the signal transduction associated with the activation of the p53–MDM2 and MAPK pathways, resulting in cell death. Resistance is an inherent part of anticancer treatment; therefore, a population of resistant cells was assessed for both drugs separately and when combined in the performed study, and its description may play a critical role in predicting and optimising the treatment response and may improve therapy scheduling [25,26]. Modelling the drug combination approach required the selection of a PD interaction parameter that could be translational in in vitro/in vivo extrapolation. The selected β parameter from the MuSyC drug interaction model was the most optimal solution for the modelling data obtained from the present study; however, further PK/PD studies on mice coupled with PBPK/PD modelling are needed to validate the optimal PD interaction parameter choice. We successfully verified the created PBPK/PD models with different dosing and scheduling regimens from external PD data. The final TGI models for the unperturbed and perturbed groups properly predicted tumour volume within 20% of the mean relative error acceptance criteria for the data from the present study and external PD data.

Among the most relevant limitations of the present work is that the current models are restricted only to the experimental data available from a single mouse study with a limited number of animals (*n* = 6); additionally, no data from other mouse melanoma xenografts were available. Furthermore, differences in the exposure ratios (AUC ratio) in the models using external PD data may have been caused by the different formulations used in those studies, but they could also be explained by the variability in the resistance parameter lambda. However, because the used cell lines were authenticated, the possibility that the genomic drift impacted the arising resistance is unlikely. The selected TGI model for the MDM2 inhibitor is similar to already reported models for RG7388 (idasanutlin) [27] and HDM201 (siremadlin) [28] molecules from the same class. Even though those models were built on data from osteosarcoma SJSA-1 derived mouse xenografts, common mechanisms such as delayed drug effects and arising resistance can be concluded for this class of small-molecule inhibitors.

The results of an in-depth analysis of the TGI models’ parameter dependencies allowed us to extrapolate PD model predictions for different doses and dosing frequencies of the studied drugs after one or two cycles of the therapy, which allowed us to select the most optimal therapeutic strategies that ensure a high efficacy. The modelling and simulation approach suggested several effective dosing schedules; however, the most important are those assuming tumour stasis and complete tumour regression. The simulations of the tumour volume after two cycles of the combined therapy at HEDs revealed that at least 40 days of continuous dosing of trametinib combined with siremadlin (regardless of the siremadlin dosing regimen) is needed for the tumour to completely regress (tumour volume ≤ 32 mm^3^). Moreover, these simulations suggested that the synergistic efficacy of the siremadlin and trametinib drug combination reduces the number of successive trametinib doses needed to achieve tumour stasis (tumour volume ≤ 170 mm^3^) to only 21 doses in the treatment cycle. This might be especially useful for patients who may develop hypersensitivity, a serious skin rash, or other adverse events grade 2 or higher after using trametinib [29]. Such suggestions may play a role in the development of a potential clinical trial protocol to study melanoma-bearing patients that will be treated with a siremadlin and trametinib combination.

In the simulations of the combination, trametinib seemed to be less effective than in monotherapy, which might be explained by the assumed higher resistance to the therapy when both drugs are simultaneously administered (but smaller parts of the cell population will be resistant). Additionally, by comparing the results from the efficacy simulations after 40 doses of trametinib (Appendix A) with data from the administration of 36 doses (Appendix A, external PD data from the literature), the simulation results suggested a higher efficiency, which might be related to the variability in the lambda parameter or exposure ratio. The presented PBPK/PD translational approach also has other associated limitations, such as not considering the influence of MDM2 inhibition in stromal or immune microenvironments [30,31,32]. Nonetheless, despite these many limitations, the developed PBPK/PD models reasonably accurately described the PK and time course of the tumour growth across all doses and dosing schedules.

Further analyses are encouraged to externally validate the developed PBPK/PD models for siremadlin, trametinib, and their combination toward predicting tumour volume after human equivalent dose administrations.

The in vitro/in vivo translational approach presented in this study facilitated the determination of the most synergistic and efficacious schedules and dose levels for the siremadlin and trametinib combination in mice and may provide a rationale for planned translational modelling between mice and melanoma-bearing patients.

Recently published clinical data on siremadlin [28,33] and trametinib [29,34] combined with the findings of this study may support the extrapolation of animal PBPK/PD data into a clinical situation. Nonetheless, due to the limited amount of in vivo drug combination data available from this study, such extrapolation may only predict the initial efficacy in patients. The performance of additional in vivo efficacy studies with a larger number of animals and different melanoma xenografts is warranted to improve simulation predictions, although the performance of virtual clinical trials (VCTs) may also facilitate simulation prediction improvement regarding the melanoma patient subpopulation. Further development of the clinical PBPK/PD models for siremadlin and trametinib is needed to construct a proper drug combination model in clinical settings.

PBPK/PD modelling is a powerful tool that allows researchers to properly estimate the in vivo efficacy of anticancer drug combinations based on the results of in vitro studies. Such an approach may indicate the most efficacious dosing strategies. This method allows for better planning of the clinical trials and estimation of drug combination efficacy in such trials on the virtual representation of cancer patients.

## 4. Materials and Methods

### 4.1. Materials

Siremadlin (catalogue number HY-18658) and trametinib (catalogue number HY-10999) used in this study were obtained from MedChemExpress. PEG 400 (catalogue number 81172) and Cremophor RH40 (catalogue number 07076) were provided by Merck (formerly Sigma-Aldrich), EtOH (catalogue number 1016/12/19) was provided by POCH, and Labrafil M1944CS (catalogue number 178290) was provided by Gattefosse. The A375 cell line used in the single drug administration in vivo studies was obtained from American Type Culture Collection (CRL-1619). For drug combination in vivo studies, the A375 cell line was provided by European Collection of Authenticated Cell Cultures (88113005).

### 4.2. Software

PK parameters were estimated with Microsoft Excel (Excel version 2016, Microsoft Corporation, Redmond, WA, USA, 2016, https://www.office.com). Digitalisation of the literature-derived data was performed with the use of WebPlotDigitizer software (version 4.4, Ankit Rohatgi, Pacifica, CA, USA, 2021, https://automeris.io/WebPlotDigitizer). PD modelling was performed with Monolix software (Monolix version 2021R1, Lixoft SAS, Antony, France, 2022, http://lixoft.com/products/monolix/). Monolix custom PD model in Mlxtran can be found in Code S1. PBPK/PD modelling was performed in Simcyp simulator software (Simcyp Animal V21, Certara UK Limited, Sheffield, UK, 2022, https://www.certara.com/software/simcyp-pbpk). Custom PK interaction and PD models in Lua can be found in Codes S2 and S3. The relationship between PD parameters in the universal PD model was determined by 2D curve fitting in Microsoft Excel and 3D curve fitting in Python ZunZun3 tool (ZunZunSite3, James R. Phillips, Birmingham, AL, USA, 2016, http://findcurves.com).

### 4.3. Studies Involving Animals

Crl:CD-1-*Foxn1^nu^* female 4–5-week-old mice from Charles River Germany inoculated subcutaneously with A375 cells were used for in vivo studies. Determination of compound concentrations in plasma, heart, liver, spleen, muscle, brain, kidney, A375 tumour, lung, gut, and skin tissue homogenates were performed with the use of a quantitative LC-MS/MS system. Tissues were resected at the following timepoints: 1.5, 4, 8, 24 h (*n* = 3 per timepoint). Pharmacokinetic parameters (AUC, C_max_, relative AUC ratio, and T_max_) and tissue:plasma partition coefficients (Kp) were calculated using MS Excel 2016. Area under the concentration versus time curve was calculated using the linear trapezoidal rule. Tissue:plasma partition coefficients (Kp) were calculated as proposed by Rodgers and Rowland [35] with an assumption of linear pharmacokinetics [36] (Equation (1)).
Kp (K_tissue:plasma_) = C_tissue,ss_/C_plasma,ss_ = AUC_tissue_/tau/AUC_plasma_/tau = AUC_tissue_/AUC_plasma_,(1)
where C is concentration in particular tissue or plasma at steady state (ss), AUC is observed area under the curve for particular tissue/plasma, and tau is the dosing interval.

Determination of tumour growth was performed after oral gavage of vehicle (60% PEG 400 (*v*/*v*), 10% Cremophor RH40 (*v*/*v*), 10% EtOH (*v*/*v*), and 20% Labrafil M1944CS (*v*/*v*)), siremadlin, trametinib, or their combination in vehicle. The volume of the administration (10 mL/kg) of the compounds was always adjusted to the mouse body weight. Initial tumour volumes, doses, dose schedules, and numbers of animals in particular in vivo studies are summarised in Table 1.

Tumour volume (V) was recorded with an electronic calliper twice or thrice a week and was calculated based on its length and width using the prolate ellipsoid equation (Equation (2)).
V = d^2^ × D/2,(2)
where d is tumour width (mm) and D is tumour length (mm). For tumour volume simulation, it was assumed that 1 cm^3^ = 1 mL.

### 4.4. Physiologically Based Pharmacokinetic Models

#### 4.4.1. General PBPK Modelling Strategy

The modelling strategy was based on “middle-out” approach combining advantages of “bottom-up” and “top-down” approaches, whereby some parameters were fixed (such as in vitro determined or literature-derived data for siremadlin and trametinib [37,38,39,40]) and others were estimated. Parameter estimation (PE) was performed using the PE Module of the Simcyp Animal V21 using the Nelder–Mead method, weighted least squares by the reciprocal of the square of the maximum observed value as the objective function, and the termination criterion defined as the improvement of less than 1% of the objective function value. Optimisation was performed manually to fit the observed data. Due to the limitations of the mouse model in the current V21 version of Simcyp Animal software, estimation of inter-individual variability was not possible; therefore, models were fitted to the averaged values of PK and PD data at particular timepoints. PBPK model performance was evaluated based on the “2-fold” criterion for maximum concentration (C_max_) and area under the concentration vs. time curve (AUC) and T_max_ [41,42]. A graphical representation of PBPK model development is presented in Appendix A.

#### 4.4.2. Mouse Population

The physiological parameters of the Simcyp mouse population were modified to reproduce the CD-1 nude mouse population used in the experimental procedure. With regard to this, body weight, cardiac output [43], tissue volumes, blood flows, and tumour properties (including tumour tissue volume, blood flow, composition, and pH) were adapted to the studied population. Since no raw data for blood flow in the A375 xenograft were available [44], blood flow was set as in human melanoma xenografts [22,23,24].

#### 4.4.3. PBPK Model Verification

The final siremadlin and trametinib PBPK models were compared with external PK data [13,14] through both visual check and numerical analysis. Experimental and predicted longitudinal plasma concentration (Cp) and tissue concentration (Ct) profiles were generated, including the mean predicted concentrations. Local sensitivity analysis (parameter scanning) was performed to evaluate the relative impact of fa, HepCL, and fu_inc in the plasma PK parameters (AUC_0–24h_ and Cmax) for siremadlin and fa and CL_iv_ for trametinib in the ranges presented in Appendix A. The performance of the siremadlin and trametinib PBPK models was assessed by the fold error for each tissue, which referred to the ratio of the predicted AUC_0–24h_, C_max_, or T_max_ to the observed AUC_0–24h_, C_max_, or T_max_ respectively (Equation (3)). AUC_0–24h_ was calculated by the linear trapezoidal rule. Both visual checks and numerical analyses were performed in Microsoft Excel 2016.
Fold Error PK parameter = Predicted PK parameter/Observed PK parameter,(3)

### 4.5. Pharmacodynamic Modelling

#### 4.5.1. General PD Modelling Strategy

Optimal PD models used further in PBPK/PD modelling were established in Monolix software. Due to the comprehensive library of Monolix PD (TGI) models [45], the first step in proper PD selection was based on the screening of various PD models with the use of an automatic model initialisation (auto-init) function. Auto-init allows finding the good initial values of parameters before starting the population modelling approach. This custom optimisation method is performed on the pooled data, without inter-individual variability, and as result finds only a local minimum of the fitted PD model (the global minimum has yet to be set). Model selections were based on visual inspection of individual observed vs. predicted data and comparisons of resulting values of model score (Equation (4)).
Model score = −2 × log-likelihood (−2LL, called also objective function value—OFV) + corrected Bayesian Information Criteria (BICc),(4)

−2LL and BICc were estimated by linearisation method to accelerate calculations. For TGI models further developed in Simcyp Animal, the goodness of TGI model fit was evaluated based on the mean relative error (RE) value (Equation (5)) being < 20% as proposed in [46]:RE (%) = 100 × (Predicted Tumour Volume − Observed Tumour Volume)/Observed Tumour Volume,(5)

#### 4.5.2. PD (TGI) Model Development and Verification

The first stage of TGI model development was focused on the selection of a mathematical model describing properly unperturbed tumour growth of A375 xenografts. Different tumour growth models assuming growth saturation such as logistic, generalised logistic, hybrid Simeoni–logistic, Gompertz, exponential Gompertz, and von Bertalanffy were evaluated and fitted to the data from the vehicle group in the Monolix software. The selected unperturbed logistic growth model assumes an exponential growth rate (kge) which decelerates linearly with respect to the initial tumour size (TS0) and is described by the following Equation (6):dTS/dt = kge × TS0 × (1 − (TotalTS/TSmax)),(6)
where kge is tumour growth (1/day), TS0 is initial tumour size (mL), TotalTS is total tumour size (mL), and TSmax is maximal tumour size (mL).

After the characterisation of tumour growth, the tumour growth inhibition models (perturbed models) were developed for siremadlin, trametinib, and their combination separately based on observed tumour volume data. Several structural TGIs were compared in terms of model score in order to select the best fit to the averaged tumour volume data.

A screen of multiple models incorporating introduction of the type of killing hypothesis (Norton–Simon or Skipper–Schabel–Wilcox log-kill), dynamics of treatment effect (linear, Emax, Emax–Hill, exponential kill), type of treatment effect delay (cell distribution model [47] or signal distribution model with 3 or 4 transit compartments [48]), type of resistance arising (claret exponential [49] or 2-population model [48,50,51,52]), and value of gamma (PD interaction parameter) (Synergyfinder-derived δ score or β parameter calculated with synergy package [6]) was performed in Monolix software.

The most important features of the selected perturbed TGI model characterising single siremadlin-, single trametinib-, and combination-treated groups are written in the following equations and initial conditions:TotalTS(t) = TS(t) + TSr(t),(7)
TS_0 = TS0,(8)
TSr_0 = TSr0,(9)
TSmax = crck × (TS0 + TSr0),(10)
kkill_Siremadlin = a × Siremadlin dose × number of doses,(11)
kkill_Trametinib = b × Trametinib dose × number of doses,(12)
kkill_combination = (kkill_Siremadlin × AUC_ratio_Siremadlin + kkill_Trametinib × AUC_ratio_Trametinib) × gamma,(13)
kkill = kkill_Siremadlin/kkill_Trametinib/kkill_combination,(14)
C(t) = C_Siremadlin/C_Trametinib/(C_Siremadlin + C_Trametinib),(15)
K(t) = kkill × (1 − e^(−s × C(t))),(16)
K1_0 = 0,(17)
K2_0 = 0,(18)
K3_0 = 0,(19)
K4_0 = 0,(20)
dK1/dt = (dK − K1)/tau,(21)
dK2/dt = (K1 − K2)/tau,(22)
dK3/dt = (K2 − K3)/tau,(23)
dK4/dt = (K3 − K4)/tau,(24)
dTS/dt = 0,(25)
dTSr/dt = 0,(26)
dTS/dt = (kge*TS*(1 − (TotalTS/TSmax))) − (K4 + ksr*K4)*TS,(27)
dTSr/dt = (kge*TSr*(1 − (TotalTS/TSmax))) + (ksr*K4*TS) − (K4/lambda*TSr),(28)
where resistance is defined by the introduction of sensitive (TS—tumour size) and resistant population (TSr—resistant tumour size) of cancer cells. It was assumed that at time 0, the total tumour size (TotalTS) is represented by the sum of TS and TSr cells (Equation (7)). TotalTS is assumed to be ~100% comprising TS cells because TSr0 = ~0 at t = 0. Initial volumes of TS and TSr at time = 0 are denoted as TS0 and TSr0 (Equations (8) and (9)). Correlation between initial tumour size (for TS0 and TSr0, respectively) and maximal tumour size (TSmax) was parametrised as the crck parameter as proposed in [28] (Equation (10)). The tumour-cell-killing constant of siremadlin or trametinib was determined to be dependent on the dose and number of doses, after introducing compound-specific killing constants—a and b (Equations (11) and (12)). The killing constant for combination (kkill_combination) was characterised as the sum of the siremadlin and trametinib killing constants adjusted with PK interaction parameters (AUC ratio parameter which was calculated for siremadlin + trametinib 100 + 1 mg/kg dose and estimated for the other doses) as well as gamma (PD interaction parameter)—β parameter determined from analysis of in vitro data (Equation (13)). Depending on the treated group, the killing constant could be assigned to the killing constants of siremadlin, trametinib, or their combination (Equation (14)). Total plasma concentration of siremadlin, trametinib, or their combination was used as the input for the drug effect (Equation (15)). The tumour growth inhibition model uses a log-kill killing hypothesis with the treatment dynamics following exponential kill kinetics: kkill is the killing constant, and s is the killing constant coefficient (Equations (16), (27), and (28)). A delay of killing effect (K) has been implemented by the introduction of 4 signal transit compartments (K1, K2, K3, K4), as suggested by [48]. The duration of this delay is determined by the parameter tau (Equations (21–24)). It was assumed that transit compartments equal 0 in time = 0 (Equations (17–20)). It was also assumed that initial change of tumour volume for sensitive and resistant cells populations equal 0 (Equations (25) and (26)). The tumour logistic growth model and growth rate (kge) were assumed to be the same for the sensitive and treatment-resistant cell populations. Acquired resistance to the therapy also assumes that part of sensitive cells will convert into resistant ones. The conversion rate from sensitive to resistant population is regulated by a rate constant denoted as ksr as previously proposed [28,53] (Equations (27) and (28)). It was assumed that studied drugs are also inducing a killing effect on the resistant cell population but with a reduced potency (Equation (28)). The parameter lambda denotes the fold-change loss in drug potency on resistant cells relative to sensitive cells. Units for particular parameters are summarised in Appendix A.

In the next stage of TGI model development, models for single drug administration were compared with external efficacy data: siremadlin efficacy from a previously performed study on mice xenografted with A375 cells (unpublished data, courtesy of Adamed Pharma) and trametinib efficacy data digitised from the literature [54,55] (studies carried out on mice xenografted with A375 cells). The applied TGI models allow the tumour volume data in external efficacy studies to be fitted properly, therefore validating those models (Appendix A). Due to differences in exposure (AUC_0–24h_) in external data, the value of the killing-effect-related parameter (kkill) was adjusted with the AUC ratio parameter which was calculated for siremadlin based on previously performed PK studies (unpublished data, courtesy of Adamed Pharma as shown in Appendix A) and was estimated in particular studies for trametinib.

In the last step, unperturbed and perturbed tumour growth inhibition models for siremadlin, trametinib, and their combination previously developed in Monolix were translated into Lua programming language and applied within the Simcyp Animal V21 for further development to achieve a mean relative error (RE) value of < 20% (Equation (5)). A graphical summary of TGI model development and verification is presented in Appendix A.

#### 4.5.3. TGI Model Parameter Dependence Estimations (Universal Model Development)

After the construction of the final PD models, relationships between input parameters were established in order to construct universal TGI models which allowed the proper prediction of tumour volume even after using different doses and dosing frequencies of the studied drugs.

Relationships between input parameters were developed for siremadlin, trametinib, and combination TGI models separately and only for selected input parameters (crck, kge, exposure ratio, kkill, and tau), while the other ones determined experimentally or estimated through TGI model development (TS0, TSr0, s, lambda, ksr, and gamma) were fixed at specific values (Appendix A).

Development of the dependency of the parameters was based on the initial assumption that values of input parameters scaled with the dose and were summed up in the combination TGI model. Various mathematical equations were screened in the ZunZun3 standard 3D equations library (including over 300 equations from bioscience, enzyme kinetics, exponential, logarithmic, polynomial, power, rational, sigmoidal, trigonometric, and many more miscellaneous equations) in order to find the best fit describing dependencies between input parameter data. Then, the best models (in terms of the lowest sum of squared absolute error) were verified on the siremadlin and trametinib verification datasets.

#### 4.5.4. Tumour Volume Simulation for Drug Combination at Human Equivalent Doses

Tumour volume for studied drugs and their combination was estimated using universal PBPK/PD model within a 0–960 h simulation timeframe at the human equivalent doses and clinically examined dosing regimens for each drug (Appendix A). For simulation purposes, the initial tumour size (TS0) was assumed to be 170 mm^3^ (0.17 mL); therefore, the cut-off for tumour stasis was set to be ≤ 170 mm^3^ at the end of the simulation (960 h). Following complete eradication of the tumour, scarring often occurs at the site of tumour implantation, leaving behind connective tissue that may be mistaken for a small tumour; thus, tumour width or length below the limit of detection (4 mm) resulting in tumour volume of 32 mm^3^ was selected as the cut-off for complete tumour regression at the end of the simulation (960 h). The chosen cut-off for complete response (tumour volume ≤ 32 mm^3^) is in line with reported values [56,57,58]. TGI model parameters for simulations of tumour volume at HEDs for studied compounds after 1 or 2 cycles of therapy are summarised in Appendix A.

Animal doses equivalent to human doses were calculated according to Equation (29) [59].
Animal equivalent dose (AED) [mg/kg] = Human equivalent dose (HED) [mg/kg]/(Weight_animal_ [kg]/Weight_human_ [kg])^(1–0.67)^,(29)
where the mean weight of mice was 0.02755 kg (mean mouse weight in current study) and human weight selected as typical patient weight was 70 kg.

## Figures and Tables

**Figure 1 ijms-23-11939-f001:**
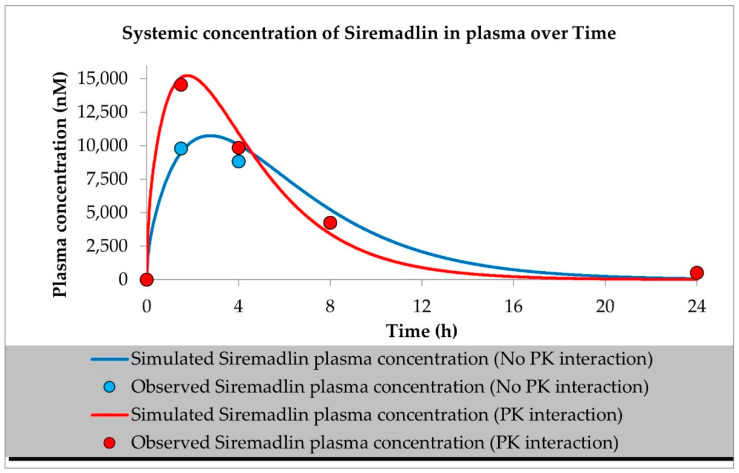
PBPK model of siremadlin with and without PK interaction. Observed data are means from *n* = 3.

**Figure 2 ijms-23-11939-f002:**
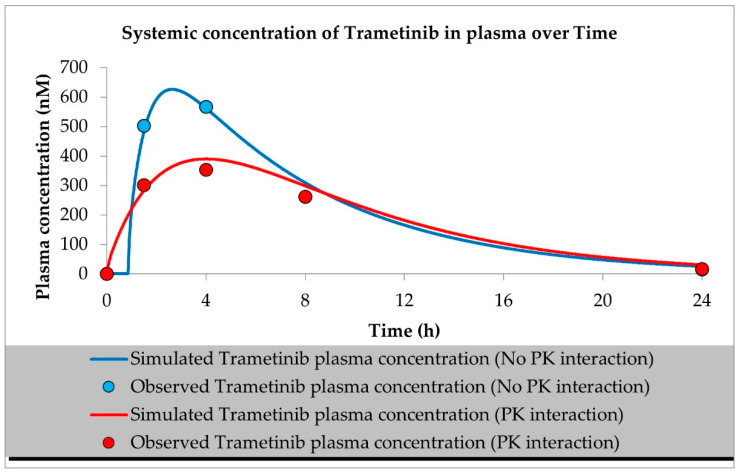
PBPK model of trametinib with and without PK interaction. Observed data are means from *n* = 3.

**Figure 3 ijms-23-11939-f003:**
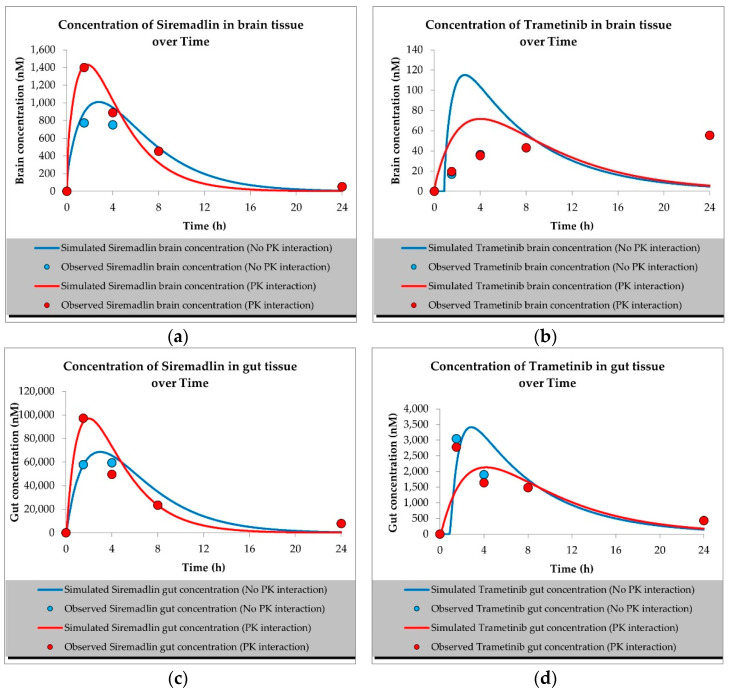
Values of predicted concentrations with and without PK interaction for (**a**) siremadlin and (**b**) trametinib in brain tissue; (**c**) siremadlin and (**d**) trametinib in gut tissue; observed data are means from *n* = 3.

**Figure 4 ijms-23-11939-f004:**
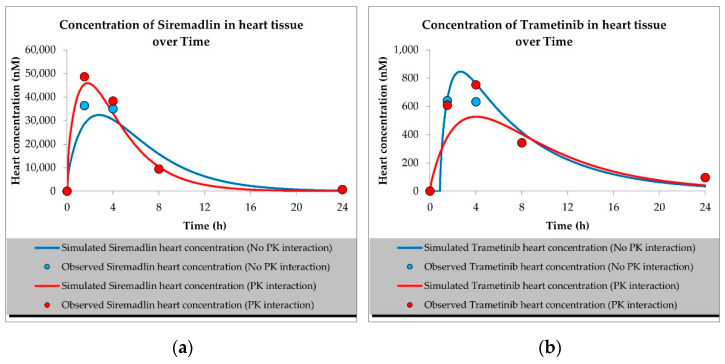
Values of predicted concentrations with and without PK interaction for (**a**) siremadlin and (**b**) trametinib in heart tissue; (**c**) siremadlin and (**d**) trametinib in kidney tissue; observed data are means from *n* = 3.

**Figure 5 ijms-23-11939-f005:**
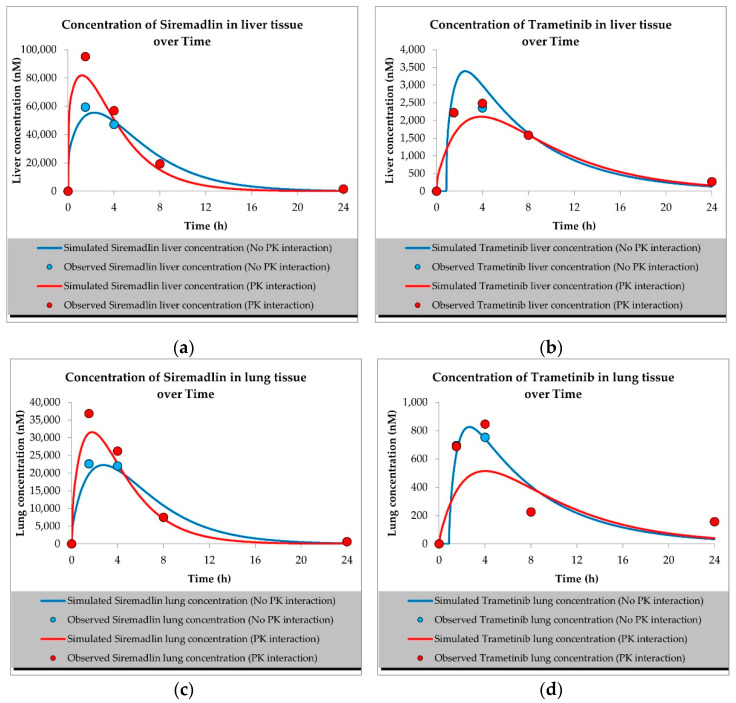
Values of predicted concentrations with and without PK interaction for (**a**) siremadlin and (**b**) trametinib in liver tissue; (**c**) siremadlin and (**d**) trametinib in lung tissue; observed data are means from *n* = 3.

**Figure 6 ijms-23-11939-f006:**
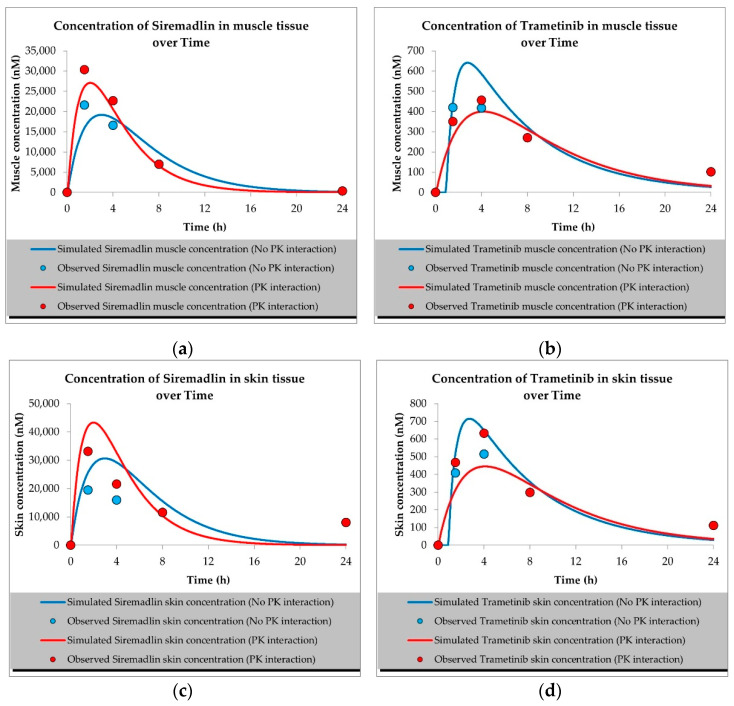
Values of predicted concentrations with and without PK interaction for (**a**) siremadlin and (**b**) trametinib in muscle tissue; (**c**) siremadlin and (**d**) trametinib in skin tissue; observed data are means from *n* = 3.

**Figure 7 ijms-23-11939-f007:**
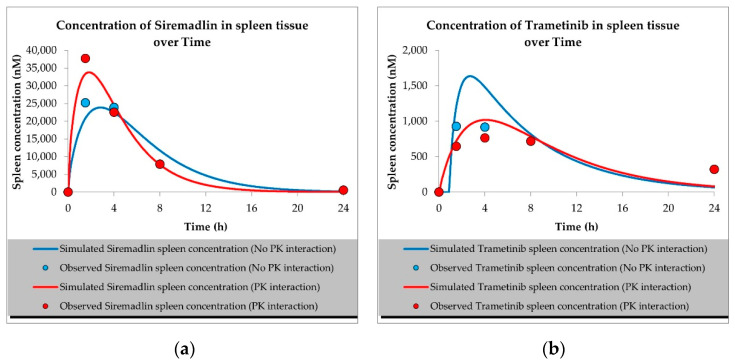
Values of predicted concentrations with and without PK interaction for (**a**) siremadlin and (**b**) trametinib in spleen tissue; (**c**) siremadlin and (**d**) trametinib in A375 tumour tissue; observed data are means from *n* = 3.

**Figure 8 ijms-23-11939-f008:**
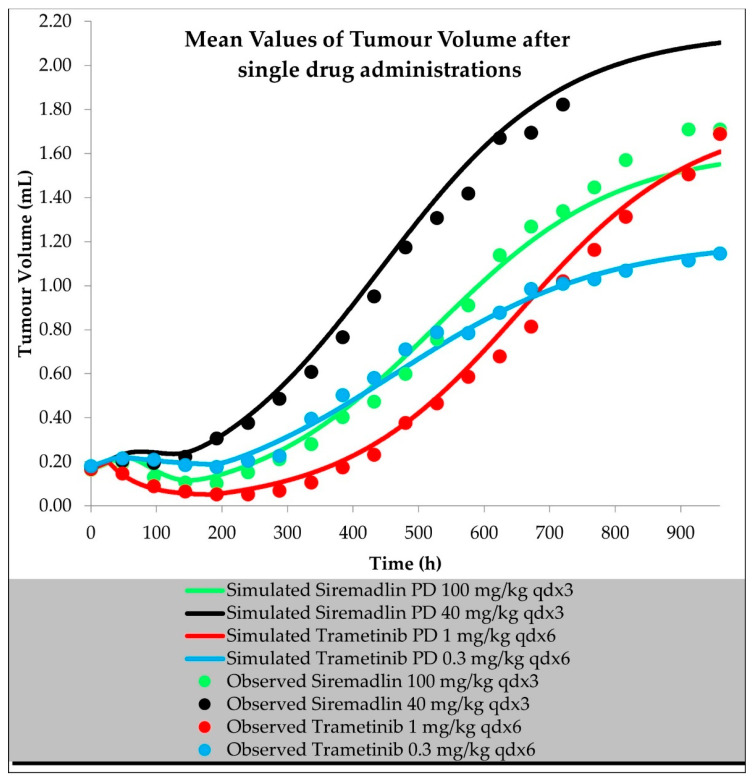
Tumour volume simulations after single drug administration of siremadlin and trametinib. Observed data are means from *n* = 6.

**Figure 9 ijms-23-11939-f009:**
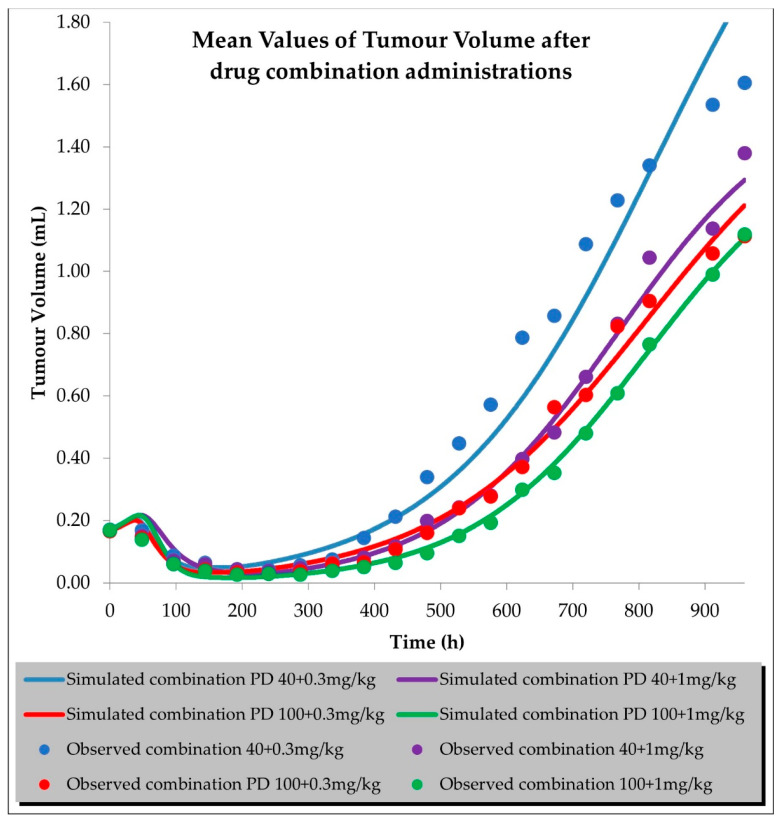
Tumour volume simulations after combined drug administration of siremadlin and trametinib. Observed data are means from *n* = 6.

**Figure 10 ijms-23-11939-f010:**
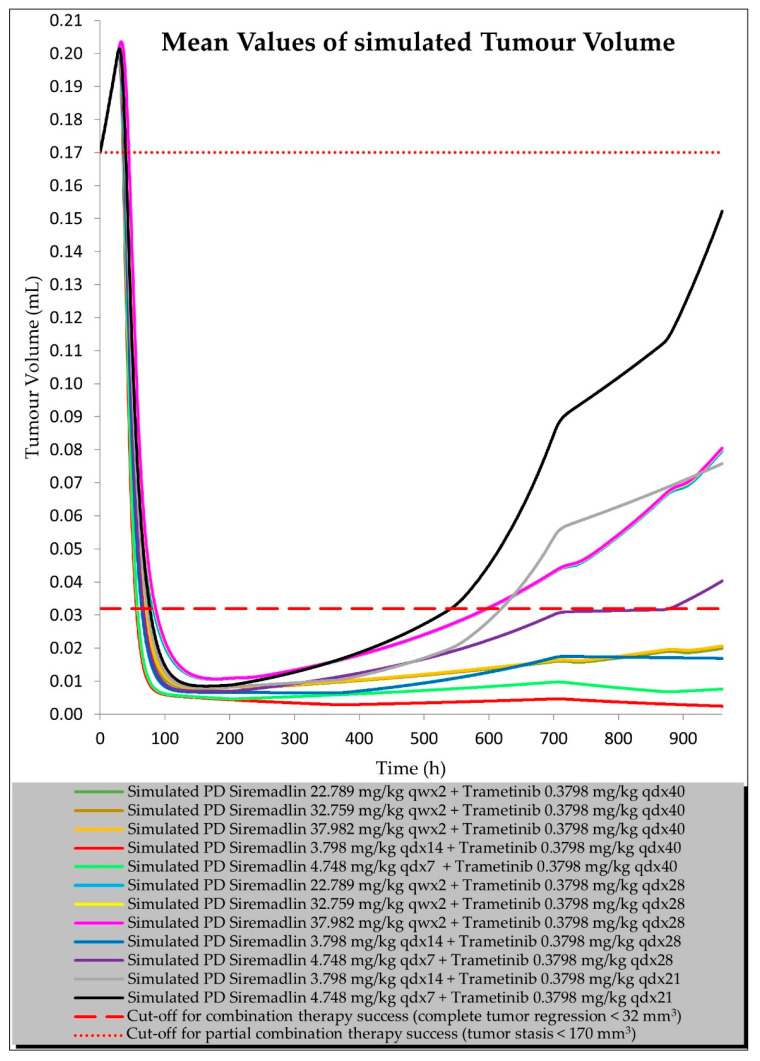
Simulation of siremadlin and trametinib combination efficacy at HEDs (2 therapy cycles) using universal TGI model.

**Table 1 ijms-23-11939-t001:** Summary of performed in vivo studies on CD-1 nude mice xenografted with A375 tumour.

Compound	Initial Tumour Volume (mm^3^)	Doses (mg/kg)	Dose Schedule	N	Comments
Vehicle (Adamed)	~135	-	q1dx5/q7dx2	10	Adamed reference
Siremadlin	~137	25/50	q1dx5	10	Adamed reference
Siremadlin	~137	50/100	q7dx2	10	Adamed reference
Vehicle (current study)	~162	-	qdx6	11	Efficacy in current study
Siremadlin	~163–172	40/100	qdx3	6	Efficacy in current study
Trametinib	~167–180	0.3/1	qdx6	6	Efficacy in current study
Siremadlin + Trametinib	~165–169	40 + 0.3/40 + 1/100 + 0.3/100 + 1	qdx3/qdx6	6	Efficacy in current study
Siremadlin	~300	100	qdx1	12	PK in current study
Trametinib	~300	1	qdx1	12	PK in current study
Siremadlin + Trametinib	~300	100 + 1	qdx1	12	PK in current study

## Data Availability

The data presented in this study are available in the article or supplementary material. Raw data from PK and PD studies are available on request from the corresponding author.

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
