# Peer review of "In Vitro/In Vivo Translation of Synergistic Combination of MDM2 and MEK Inhibitors in Melanoma Using PBPK/PD Modelling: Part II"

_ijms, 2022, doi:10.3390/ijms231911939_

Round 1

Reviewer 1 Report

I will put comments below in bullet point form:

- The introduction seems quite short, and I believe only 2 papers are cited. Especially as one of the two papers cited is by the authors, and not yet published. Polytherapy is not a novel field, and there are many papers that could be cited here. A single and obvious example here is the authors stating on L53-54 " the lack of consensus on which theoretical model should be used", this seems an obvious point that requires citations and expansion to explain the lack of consensus. 

- For the results section, I am a little unclear why so much of the data is in the supplemental section. Particularly for the other tissues analyzed (S1-10), some of these need to be in the main figures and then can be more easily assessed by readers. Also, it would be helpful if the authors could attach to the legend for each supplementary table, which figure the data refers to. 

-Figure 1-2 and Fig S1-10 are referenced to a single sentence at the start of the results (and subsequently), which makes it challenging to assess the validity of the claims. I think the authors need to re-write and expand out this section, breaking down the analysis of individual figures more than is currently done.

- Why did the authors choose 2-fold error, rather than the more standard 1.5-fold error? I'm not convinced for the other tissues placed in the supplementary figures that the prediction does adequately describe the observed values, and certainly that plasma levels were representative of other tissues. The authors state that there were exceptions that did not fit within this error level (L91-92).

- Only two data points were taken for the drugs when injected in isolation? Is this correct? If so, why? How do we assess observed Tmax values with two data points? And in the Table S4, the observed Tmax for the brain is at 24hrs? I cannot see a datapoint here. 

- Figure 3, the colour coding of the observed data is confusing and unnecessarily similar, please change so they can be more easily distinguished (as was already done in Figure 4).

- Methods, was any ethical approval for the in vivo experiments sought/required? I cannot see any mention of this? 

- Methods, it appears like the A395 cell line had two independent sources for different experiments. Can the authors confirm if this is correct, and if so, why this has occurred?  

Author Response

Dear reviewer,

Thank you very much for your valuable comments and careful review of our work. We prepared a revised version of the manuscript which has undergone English language editing by MDPI (changed fragments are marked in red). Please find our point-by-point response in red down below:

-The introduction seems quite short, and I believe only 2 papers are cited. Especially as one of the two papers cited is by the authors, and not yet published. Polytherapy is not a novel field, and there are many papers that could be cited here. A single and obvious example here is the authors stating on L53-54 " the lack of consensus on which theoretical model should be used", this seems an obvious point that requires citations and expansion to explain the lack of consensus. 

That is a very good point which is addressed in the revised version of the manuscript by expanding the introduction and cited literature. Regarding the lines 53-54 this lack of consensus is discussed in more detail in previous part of the publication cycle (please see it attached with English language editing certificate for the revised manuscript in the last page)

- For the results section, I am a little unclear why so much of the data is in the supplemental section. Particularly for the other tissues analyzed (S1-10), some of these need to be in the main figures and then can be more easily assessed by readers. Also, it would be helpful if the authors could attach to the legend for each supplementary table, which figure the data refers to. 

We much appreciate these tips regarding results. We must admit that we were concerned with the length of the manuscript (19 pages) which could be uneasy to read. That is why we wanted to shorten amount of the presented data in main text and store a lot of data in the supplementary materials. This is also a reason why for e.g. introduction or results sections were not so extensively developed/described. In the corrected version of the manuscript we added figures related to the concentrations in particular tissues to the main text. In supplemental data to the description of the supplementary tables, we added information about which figures are related to.

-Figure 1-2 and Fig S1-10 are referenced to a single sentence at the start of the results (and subsequently), which makes it challenging to assess the validity of the claims. I think the authors need to re-write and expand out this section, breaking down the analysis of individual figures more than is currently done.

Thank you for this comment. In the revised version of the manuscript mentioned figures are discussed in more detail.

- Why did the authors choose 2-fold error, rather than the more standard 1.5-fold error? I'm not convinced for the other tissues placed in the supplementary figures that the prediction does adequately describe the observed values, and certainly that plasma levels were representative of other tissues. The authors state that there were exceptions that did not fit within this error level (L91-92).

Thank you for noticing that this is very valuable comment. The reason why we have chosen 2-fold error is that it is commonly accepted criteria for PBPK modelling when evaluating the accuracy and acceptability of predictions. Moreover, such criteria are applied in practice by majority of pharmaceutical companies for regulatory submissions (please see doi: 10.1002/cpt.1013). Exceptions mentioned in lines 91-92 are generally related to the Tmax values. The reasons why those values might be mispredicted were discussed in the manuscript in the following lines (lines 108-112 and 149-151 in revised manuscript). Nevertheless, we agree that when looking at the graphs, one gets the impression that this simulation does not adequately describe the observed values for some tissues. Therefore, in the revised manuscript sentences indicating that calculated AUC, Cmax and Tmax parameters fold errors were within the 2-fold error range (0.5-2.0) were modified to be clear and indicate that fold errors for those PK parameters were in most cases in the 2-fold range.

- Only two data points were taken for the drugs when injected in isolation? Is this correct? If so, why? How do we assess observed Tmax values with two data points? And in the Table S4, the observed Tmax for the brain is at 24hrs? I cannot see a datapoint here. 

Thank you for this comment. Unfortunately, only 4 timepoints were taken into account in the PK analysis. The conducted PK/PD experiment was enormous and was involving ~200 mice at once (there were many experimental groups – 26 in total, also with compounds developed by Adamed Pharma, unpublished data). Due to number of animals and the complexity of the experiment it was decided to investigate only limited number of timepoints. As discussed in the manuscript due to sparse sampling of observed data, calculated Tmax values were reported on highest observed Cmax and therefore might be not properly determined. Regarding the observed Tmax in brain, the last observed value is at 24h and it can be found in Figure 3a and 3b. For Trametinib observed Tmax=24h and in case of Siremadlin it is lower 1.5h.

- Figure 3, the colour coding of the observed data is confusing and unnecessarily similar, please change so they can be more easily distinguished (as was already done in Figure 4).

Thank you for the suggestion. Please see the changed colours in this figure in the revised version of the manuscript (in revised manuscript figures numeration was changed, now it is Figure 8).

- Methods, was any ethical approval for the in vivo experiments sought/required? I cannot see any mention of this? 

Yes of course, there was an ethical approval which was obtained before the start of in vivo experiments. We realized that this section is missing a couple of days after we submitted this manuscript when our editor asked us about it. We sent the information that this section was intended to be filled with the following statement: “The animal study protocols were approved by the Ethics Committee of II Local Ethics Committee for Animal Experiments in Warsaw (permission for the experimental use of animals WAW2_6/2016 approved 16.03.2016 and WAW2_19/2016 approved 18.05.2016) for studies involving animals.” This information is added to the corrected manuscript.

- Methods, it appears like the A375 cell line had two independent sources for different experiments. Can the authors confirm if this is correct, and if so, why this has occurred?  

Thank you for noticing this. Yes, we can confirm that A375 cell line had two independent sources for different experiments. The reason for this was the availability of the cell line at individual suppliers for the University where the experiment was conducted.

Reviewer 2 Report

The manuscript reports the possible use of PBPK/PD modelling as a tool to understand the in vitro – in vivo extrapolation of two drugs to predict the dosing strategies. The following comments need to be addressed before considering for publication:

Major comments:

Abstract:

1) The Background section of the abstract sounds like an incomplete statement and needs revision.

"Development of method allowing to translate in vitro observed synergy between 11 HDM201 and Trametinib into in vivo synergistic efficacy between these molecules in mice model to determine the most effective therapeutic options."

2) It seems that the authors want to imply this "In the presented study, models were built on in vitro ADME and in vivo PK/PD data determined from literature or estimated in the Simcyp Animal Simulator (V21)". Please correct accordingly.

3) "in vitro determined efficacy interaction" - this phrase needs to be reframed for clarity like "interaction at the absorption and tumour disposition levels"

4) I am not sure whether Part I and Part II of this study, if accepted, will be published and available for the reader in one edition. In any regard, the opening statement, "Preclinical evidence suggests that Siremadlin and Trametinib act synergistically in the treatment of melanoma", does not introduce any new reader about the nature of the drugs (like what inhibitors they are). It would be ideal to mention a phrase or line about the description of each drug.

5) In line with the above comment, the title of the study is so broad mentioning "the synergistic combination of MDM2 and MEK inhibitors", but the study does not deal with many or all of the MDM2 and MEK inhibitors, but only one of each category. So, the title ideally should  directly mention the name of the drugs rather than their category.

6) Both Siremadlin and HDM201 have been used interchangeably in the manuscript, which is confusing to read and needs to be fixed with only one term for describing that drug.

7) Lines 75-84 have unclear sentences and descriptions. For ex: 

a) "For HDM201 and Trametinib, slightly under-estimated...." - a comma makes the sentence understandable

b) The figures or tables corresponding to the description of each statement should be indicated within brackets to facilitate the readability for a reader.

c) "Calculated AUC0-24h, Cmax and Tmax parameters fold errors" very compound noun structure unclear to understand.

The above suggestions should be incorporated for all the results sections.

8) The manuscript needs a thorough language editing as many sentences were unclear with language and grammar errors:

For ex: "In the model accounting for PK interaction, key pharmacokinetic parameters..." and "Concerning A375 tumour and brain tissues, high Tmax fold error might be related to..." are the two best examples that the missing of a comma leaves the sentences incomprehensible.

Another example is, the title of Figure 1 should be ideally "Systemic concentration of HDM201 in plasma over time".

Line 107: It should be either "model predictions" or ""models prediction"

9) Lines 93-98 are so complex and need to be split to make the sentences understandable.

10) Figures 1 and 2 should not have any decimal units in both axes.

11) The authors have performed this study based on the experimental data available from a single mice study with a limited number of animals (N=6). How can a "simulation" based study based on one single study could be "believed" to be ideal to conclude the extrapolation studies on other animal models or clinical studies in humans?

13) The manuscript needs a language editing service if considered for publication.

Lines 112-113: "Final PBPK models parameters with and without PK interaction are summarized in Tables S2 and S7-S8" should be  "The final parameters of PBPK models without and with PK interaction are summarized in Tables S2 and S7-S8".

Please note the use of "without" before "with" in this sentence which corresponds to the descriptions in the figures. The figure legends also should be changed in this order.

Lines 287-291 - even though they are concluding lines, they do not have any clear meaning with complex sentences without any comma or period.

Minor:

1) "based on in vitro studies results" should be "based on the results of in vitro studies"

2) "link it to efficacy (pharmacodynamic effect)" should be "link it to the drug's efficacy (pharmacodynamic effect)"

3) "it enables estimation of drug exposure at the site of ITS action"

4) Line 56-57, "were selected to be translational in vitro/in vivo PD interaction parameter" - there is a noun-plural disagreement that needs to be fixed.

5) Table S2 - what does "introduction interaction at the PK level" mean?

Author Response

Dear reviewer,

Thank you very much for your valuable comments and careful review of our work. We prepared a revised version of the manuscript which has undergone English language editing by MDPI (changed fragments are marked in red). Please find our point-by-point response in red down below:

1) The Background section of the abstract sounds like an incomplete statement and needs revision.

"Development of method allowing to translate in vitro observed synergy between HDM201 and Trametinib into in vivo synergistic efficacy between these molecules in mice model to determine the most effective therapeutic options."

Thank you for this advice. In the revised manuscript this section was changed to: “The development of in vitro/in vivo translational methods for synergistically acting drug combinations is needed to identify the most effective therapeutic strategies.”

 2) It seems that the authors want to imply this "In the presented study, models were built on in vitro ADME and in vivo PK/PD data determined from literature or estimated in the Simcyp Animal Simulator (V21)". Please correct accordingly.

Thank you for noticing that. In the revised manuscript it was corrected.

3) "in vitro determined efficacy interaction" - this phrase needs to be reframed for clarity like "interaction at the absorption and tumour disposition levels"

Thank you for noticing that. In the revised manuscript those phrases have been separated from previous line and reworded for clarity: “The interaction at the PK level was described by an interplay between absorption and tumour disposition levels whereas the PD interaction was based on the in vitro results.”

4) I am not sure whether Part I and Part II of this study, if accepted, will be published and available for the reader in one edition. In any regard, the opening statement, "Preclinical evidence suggests that Siremadlin and Trametinib act synergistically in the treatment of melanoma", does not introduce any new reader about the nature of the drugs (like what inhibitors they are). It would be ideal to mention a phrase or line about the description of each drug.

Both parts of publication cycle (Part I and Part II) were co-submitted to the Special Issue "Novel Strategies in the Development of New Therapies, Drug Substances and Drug Carriers 2.0" of IJMS. Part I is already after review. We will ask editors to ensure that Part I will be available for the readers in one edition. In the revised manuscript a line with description of each drug was added before previous opening statement: “One of the novel therapeutic options is the drug combination of siremadlin (MDM2 inhibitor) and trametinib (MEK inhibitor).”

5) In line with the above comment, the title of the study is so broad mentioning "the synergistic combination of MDM2 and MEK inhibitors", but the study does not deal with many or all of the MDM2 and MEK inhibitors, but only one of each category. So, the title ideally should  directly mention the name of the drugs rather than their category.

Thank you for this suggestion. Results of the original study (including other MDM2 inhibitors, data not shown) and studies reported in the literature for various MDM2 and MEK inhibitors suggest synergistic interaction (please see doi: 10.3390/cancers11010003, 10.18632/oncotarget.1918, 10.3390/cancers12082253, 10.1038/onc.2017.258, 10.1038/s41598-022-05193-z and WO2015070224). Moreover, this synergistic drug interaction for those classes of inhibitors was characterized at the molecular level. Therefore, it could be assumed that almost every MDM2 inhibitor combined with MEK inhibitor will interact with each other synergistically. This is the reason why the title of the current report is broad and refers to the synergistic interaction between MDM2 and MEK inhibitors combination.   

6) Both Siremadlin and HDM201 have been used interchangeably in the manuscript, which is confusing to read and needs to be fixed with only one term for describing that drug.

At the preclinical development stage this molecule name was encoded as HDM201 but when it entered clinical trials it was renamed to Siremadlin. In revised version of the manuscript the name of this molecule in text and figures was unified to "siremadlin". 

7) Lines 75-84 have unclear sentences and descriptions. For ex: 

  1. a) "For HDM201 and Trametinib, slightly under-estimated...." - a comma makes the sentence understandable
  2. b) The figures or tables corresponding to the description of each statement should be indicated within brackets to facilitate the readability for a reader.
  3. c) "Calculated AUC0-24h, Cmax and Tmax parameters fold errors" very compound noun structure unclear to understand. Corrected to “Calculated fold errors for AUC0-24h, Cmax and Tmax parameters were mainly within the 2-fold error range (0.5-2.0).”

The above suggestions should be incorporated for all the results sections.

Thank you for those suggestions! In the revised version of the manuscript those sentences and related article sections were corrected.

8) The manuscript needs a thorough language editing as many sentences were unclear with language and grammar errors:

For ex: "In the model accounting for PK interaction, key pharmacokinetic parameters..." and "Concerning A375 tumour and brain tissues, high Tmax fold error might be related to..." are the two best examples that the missing of a comma leaves the sentences incomprehensible.

Another example is, the title of Figure 1 should be ideally "Systemic concentration of HDM201 in plasma over time".

Line 107: It should be either "model predictions" or ""models prediction"

Thank you for paying attention to those lines and figures! In the revised version of the manuscript missing commas and descriptions of the figures in the main text and supplementary materials were corrected.

9) Lines 93-98 are so complex and need to be split to make the sentences understandable.

That is right. In the revised version of the manuscript those sentences were divided.

10) Figures 1 and 2 should not have any decimal units in both axes.

Thank you for noticing that. In the revised version of the manuscript figures do not have any decimal units in both axes when not necessary.

11) The authors have performed this study based on the experimental data available from a single mice study with a limited number of animals (N=6). How can a "simulation" based study based on one single study could be "believed" to be ideal to conclude the extrapolation studies on other animal models or clinical studies in humans?

That is a very good point, thank you! Yes, we agree that the extrapolation of human efficacy based on single mice study with limited number of animals might lead to some concerns related to the simulation of the efficacy in human (please see lines 325-328 in revised manuscript). We are aware that much more in vitro/in vivo data is needed to provide reliable simulation however the main aim of this publication cycle is to provide a method/tools that are needed for in vitro-in vivo data translation with initial human extrapolation. However, this is further addressed in revised manuscript by modifying the concluding sentences (lines 376-382 in revised manuscript): “Nonetheless, due to the limited amount of in vivo drug combination data available from this study, such extrapolation may only predict the initial efficacy in patients. The performance of additional in vivo efficacy studies with a larger number of animals and different melanoma xenografts is warranted to improve simulation predictions, although the performance of virtual clinical trials (VCTs) may also facilitate simulation predictions improvement regarding the melanoma patient subpopulation.”

13) The manuscript needs a language editing service if considered for publication.

Lines 112-113: "Final PBPK models parameters with and without PK interaction are summarized in Tables S2 and S7-S8" should be  "The final parameters of PBPK models without and with PK interaction are summarized in Tables S2 and S7-S8".

Please note the use of "without" before "with" in this sentence which corresponds to the descriptions in the figures. The figure legends also should be changed in this order.

Lines 287-291 - even though they are concluding lines, they do not have any clear meaning with complex sentences without any comma or period.

Thank you very much for those suggestions. The revised version of manuscript is after MDPI English language editing (please find English editing certificate attached). 

Minor:

1) "based on in vitro studies results" should be "based on the results of in vitro studies"

In the revised version of the manuscript this phrase was corrected.

2) "link it to efficacy (pharmacodynamic effect)" should be "link it to the drug's efficacy (pharmacodynamic effect)"
In the revised version of the manuscript this phrase was corrected.

3) "it enables estimation of drug exposure at the site of ITS action"

In the revised version of the manuscript this phrase was corrected.

4) Line 56-57, "were selected to be translational in vitro/in vivo PD interaction parameter" - there is a noun-plural disagreement that needs to be fixed.

In the revised version of the manuscript this sentence was reworded for clarity: “we chose the previously proposed synergy metrics δ score (from the Synergyfinder package analysis) and β parameter (from the Synergy package analysis) to be tested as translational in vitro/in vivo PD interaction parameters.” 

5) Table S2 - what does "introduction interaction at the PK level" mean?

Thank you for this question. Interaction at PK level mean introduction of changes in parameters related to the absorption and tumour distribution. In the revised manuscript supplementary materials this phrase was changed to “model parameters with and without interaction at PK level.” to improve clarity.

Round 2

Reviewer 2 Report

The revision addresses the comments and the manuscript is now suitable for publication.